# Influence of Acid Mine Drainage Leakage from Tailings Ponds on the Soil Quality of Desert Steppe in the Northwest Arid Region of China

**Jianfei Shi** [1,2] **, Wenting Qian** [1] **, Zhibin Zhou** [1] **, Zhengzhong Jin** [1,*] **and Xinwen Xu** [1]

[1] Xinjiang Institute of Ecology and Geography, National Engineering Technology Research Center for Desert-Oasis Ecological Construction, Urumqi 830054, China

[2] University of Chinese Academy of Sciences, Beijing 100049, China

[*] Correspondence: jinzz@ms.xjb.ac.cn

**Abstract:** As decision-making tools helping to improve the understanding of soil quality, soil quality assessment and heavy metal pollution assessment are very important for the remediation of heavy metal soil pollution. In the past, soil quality and heavy metal pollution have been studied separately, and few studies have combined them. The desert steppe in the Northwest Arid Region is an important pasture resource in China, and its soil safety has always been the focus of attention. Therefore, to understand the impact of tailing stockpiles on the soil quality of desert steppe, this study analyzed 18 indicators in the sample and analyzed the soil quality status of desert steppe based on the soil quality index (SQI) and Nemerow pollution index ($P_{com}$). The main conclusions are as follows. (1) The evaluation results of heavy metal soil pollution show that the heavy metals Cu, Ni, Cr and Cd are significant polluters, Mn is a moderate polluter and Zn is a slight polluter. The results of the positive matrix factorization model show that Cu and Ni come from industrial sources; Cr, Cd and Zn come from industrial and traffic sources; and Mn comes from natural sources. (2) Regarding the study area, the generated minimum data set contains clay, pH, soil organic matter, available phosphorus, urease and neutral phosphatase. (3) The results of the SQI show that the soil in the study area is grade V (SQI-TDS$_{ave}$ (total data set) = 0.42; SQI-MDS$_{ave}$ (minimum data set) = 0.39), and the soil condition is very poor. 4) The linear fitting results show that the SQI-MDS was positively correlated with the SQI-TDS ($R^2 = 0.79$), and SQI-MDS and SQI-TDS were negatively correlated with the $P_{com}$ ($R^2 > 0.6$). Therefore, the leakage of acid mine drainage from tailings pond accumulation has led to a significant decline in the soil quality of this desert steppe, and effective ecological restoration measures are urgently needed to ensure the sustainable stability of the steppe ecosystem.

**Keywords:** acid mine drainage; heavy metal pollution; soil quality evaluation; minimum data set; desert steppe





## 1. Introduction

Mining can provide materials and energy for the production of large-scale global market products, and promote the development of China's economy [1,2]. However, long-term large-scale mining activities not only cause local steppe degradation, vegetation damage, soil erosion and surface subsidence but also produce a large amount of mine waste, namely tailings [2]. According to statistics, there are about 12,000 tailings ponds in China, with a total tailings accumulation of more than 10 billion tons and an annual growth rate of 600 million tons [3]. Tailings ponds are a significant source of heavy metal pollution, and the heavy metals contained therein can be directly released into the surrounding environment through acid mine drainage leaks [4,5]. In some tailings ponds, curtain grouting is implemented to stop acid mine drainage leakage, but due to deficiencies in current curtain grouting technology, there is still a risk of acid mine wastewater leakage from tailings ponds, which poses a serious threat to regional ecological safety [6]. With

the transfer of China's mining center from Central China to Northwest China, heavy metal pollution caused by mining activities has further threatened the fragile desert steppe ecosystem in Northwest China [7–9]. The increase in heavy metal pollution in soil will lead to a decline in soil nutrients (total nitrogen, total phosphorus, available phosphorus, etc.) and a decrease in soil enzyme activity, which will have a negative impact on soil quality [10–12]. Xu et al. [9] showed that the higher the degree of heavy metal pollution, the lower the soil quality of desert steppes. However, there are few studies on the response of soil quality to heavy metal pollution in desert steppes in arid areas. Therefore, it is necessary to further study the relationship between desert steppe soil quality and soil heavy metal pollution, which is of great significance to understanding the changes in desert steppe soil quality, preventing desert steppe soil degradation and conserving the desert steppe ecosystem.

Soil quality is defined as the ability of a specific type of soil to function within the boundaries of natural or managed ecosystems to maintain plant and animal productivity, maintain or improve water and air quality, and support human health and habitation [13,14]. Soil quality affects the basic functions of soil, including soil water movement and plant nutrient supply, nutrient cycling and resistance to organic and inorganic pollutants [15,16]. As the most sensitive soil index, soil quality can effectively reveal the dynamics of soil conditions to reflect the impact of natural factors and human activities on soil [17]. Effective evaluation of soil quality can improve soil productivity, maintain soil sustainability and play a positive role in promoting soil ecological balance and protecting human health [18]. At present, the research on soil quality assessment is mainly focused on agricultural land [15,19,20]. There are few soil quality assessments for desert steppe, and soil physical and chemical properties are mainly selected as evaluation indicators [17,18], while biological characteristics indicators are less frequently considered, especially soil enzyme activity [21,22].

Many methods and models are used to evaluate soil quality [19,22–24]. The soil quality index (SQI) is a commonly used method that integrates different soil indicators into a simplified format, giving it more advantages than other methods [25,26]. The SQI evaluates a large number of physical, chemical and biological characteristics [27,28], but the information contained in these data may have a lot of redundancy, and the determination of a large number of indicators is expensive, so it is not feasible to take all indicators into account [29]. Larson et al. [30] proposed a minimum data set (MDS), which can not only reduce data redundancy but also generate a large amount of information with less manpower and cost. In addition, using principal component analysis (PCA) to establish the MDS can generate the weights of all indicators, avoiding the subjective impact of human factors on the results of soil quality evaluation [31]. In conclusion, the purpose of this study is to (1) establish an MDS with appropriate indicators for the evaluation of soil quality; (2) understand the status of heavy metal pollution in soil and its sources and provide a basis for the treatment of heavy metal pollution; and (3) explore the response relationship between soil quality and heavy metal pollution.

## 2. Materials and Methods

### 2.1. Site Description and Soil Sampling

The research area is located in a copper-nickel mine tailings pond (86°39′58″ E, 46°44′10″ N) in the north of Xinjiang Province and the east of Fuyun County (Figure 1). The arid and semiarid climate zone covering the temperate climate zone in the region is mainly affected by the mid-latitude westerlies throughout the year and the Siberian high-pressure system in winter [32–34]. The annual average temperature in this area is 4.1 °C, the annual average precipitation is 217.1 mm, the annual average relative humidity is 58%, the annual average evaporation is 1743 mm and the altitude is 317–3863 m [33,35]. The deposits along the Fuyun fault consist mainly of Eocene conglomerates and sands and Quaternary alluvium covering the Paleozoic metamorphic rocks and Mesozoic granitic and sedimentary rocks, and the water system mainly includes two major water systems, the Irtysh River and the Ulungur River, with an annual runoff of $4.35 \times 10^9$ m$^3$ [35,36].

The main soil types in the study area are chestnut soil and brown calcium soil [37]. The most common plants are *Seriphidium kaschgaricum* (Krasch.) Poljak., *Suaeda glauca* (Bunge) Bunge, *Polygonum aviculare* L. and *Limonium sinense* (Girard) Kuntze. Tailings pond dams are mainly constructed using mine waste rock, and tailings sand is disposed of using the traditional wet drainage method [38]. Because of ineffective curtain grouting technology, the wastewater conveying tailings sand leaks continuously, and the mine waste rock used for dam construction produces a large amount of AMD under the combined action of water, air and microbial activities, which has a serious impact on the desert steppe and poses a large ecological safety hazard. To carry out an ecological remediation test for heavy metals in the soil in this AMD-contaminated area, we investigated and sampled the soil in the remediation test area. The repair test area was located in the northwest of the tailings pond, with an area of about $1 \times 10^5$ m$^2$. The sampling method used was random sampling, and the coordinates of the sample points were determined using a hand-held GPS instrument. The sampling depth of soil samples was 0–20 cm, and there were 30 soil samples in total.

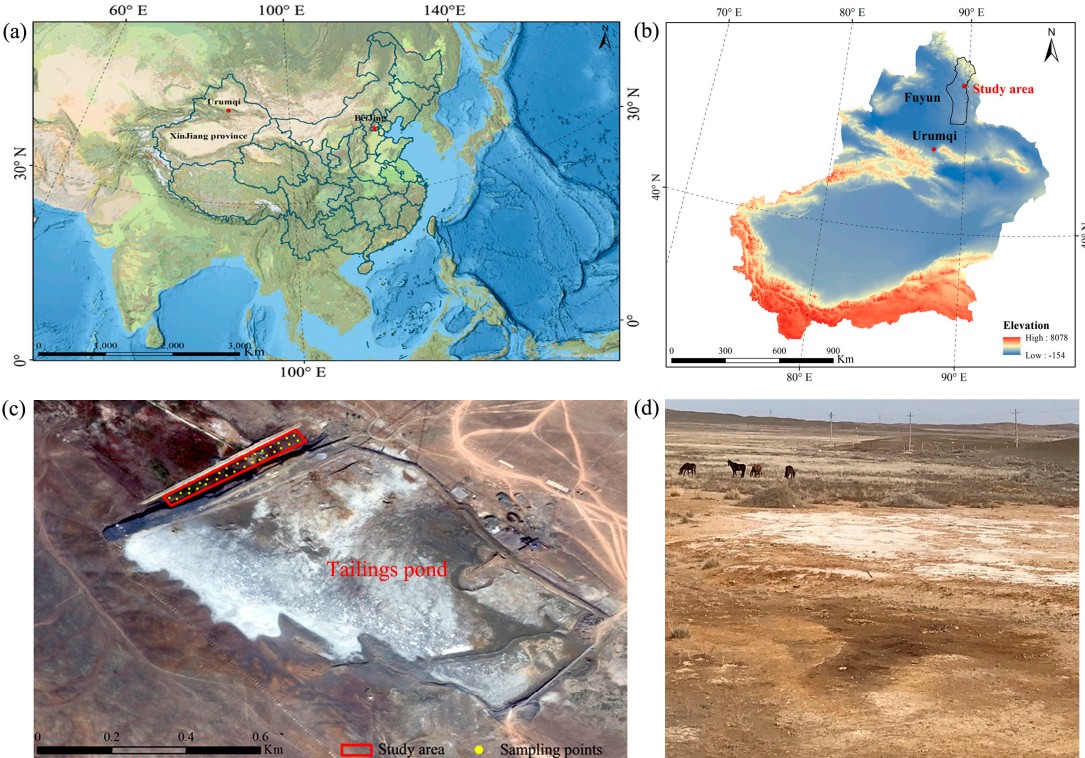

**Figure 1.** Study area and sampling points: (**a**) The location of Xinjiang in China; (**b**) The location of Fuyun County in Xinjiang; (**c**) Distribution of sampling points; (**d**) Pictures of the contaminated area. (The map is based on the standard map number GS (2016) 1666 downloaded from the standard map service website of the National Bureau of Mapping Geographic Information, and the base map is not modified. https://earthexplorer.usgs.gov/, accessed on 15 November 2022).

### 2.2. Analysis of Soil Samples

The soil samples were placed indoors for air drying. After removing gravel and plant residues, they were ground and screened with 2 mm and 0.75 mm nylon sieves. The percentage of clay, silt and sand in the soil was determined using a laser particle sizer [39]. Soil water content (SWC) was determined using a drying method [40]. The conductivity (EC) of aqueous soil extract was measured with a conductivity meter [41]. The pH value of aqueous soil extract was determined with a pH meter [42]. Soil organic matter (SOM) was determined using the Walkley–Black method [43]. Total nitrogen (TN) was determined using the Kjeldahl digestion method [44] and the alkaline hydrolysis diffusion method to determine alkali-hydrolyzable nitrogen (AN) [45]. After digesting the sample, total phos-

phorus (TP) was determined via spectrophotometry [46]. Available phosphorus (AP) was determined using 0.5 mol $L^{-1}$ sodium bicarbonate solution and a spectrophotometer [46]. Total potassium (TK) was detected via sodium hydroxide melting flame photometry [47]. Available potassium (AK) can be determined using 1 mol $L^{-1}$ ammonium acetate solution and detection with a flame photometer [47]. The soil enzyme activity was determined according to the method of Guan [48]. Urease (URE) was determined using sodium phenol sodium hypochlorite colorimetry. Invertase (INV) was determined using 3,5-dinitrosalicylic acid colorimetry. Neutral phosphatase (NPH) was determined using sodium phenylene phosphate colorimetry. Polyphenol oxidase (PPO) and catalase (CAT) were determined via iodometric titration. To determine the concentration of heavy metals, the soil samples were completely digested with $HNO_3$-HF-HCl ($HNO_3$:HF:HCl = 3:1:1, volume ratio), and the trace elements (Cu, Ni, Cr, Mn, Zn and Cd) were analyzed using ICP-OES, and a blank control, duplicate samples and reference materials (GSS-25) were used for quality control. The recovery extent of each element was between 92% and 104%, and the results met the requirements of quality control.

*2.3. Assessment of Soil Pollution*

2.3.1. Single-Factor Pollution Index Method

The single-factor pollution index ($P_i$) most directly reflects the level of pollution from environment indicators; the calculation formula is as follows [49,50]:

$$P_i = C_i / S_i. \tag{1}$$

$P_i$: single-component contamination index, $C_i$: measured concentration of examined metal $i$ in the soil and $S_i$: background concentration of metal $i$. Because different soil parent materials occur in different regions, heavy metals have a high spatial heterogeneity. Therefore, the background value of heavy metals in Xinjiang soil was selected as the standard value in the assessment of soil heavy metal pollution in this study. The evaluation results are divided into five grades: $P_i \leq 0.7$, safe; $0.7 < P_i \leq 1.0$, warning; $1 < P_i \leq 2$, slight pollution; $2 < P_i \leq 3$, moderate pollution; and $P_i > 3$, heavy metal pollution.

2.3.2. Nemerow Comprehensive Pollution Index Method

The Nemerow composite index ($P_{com}$) method not only takes into account all the individual evaluation factors but also highlights the importance of the most contaminating element. The calculation formula is as follows [50,51]:

$$P_{com} = \sqrt{\frac{P_{max}^2 + P_{ave}^2}{2}}. \tag{2}$$

$P_{com}$: composite contamination index, $P_{ave}$: average value of the single-factor index and $P_{max}$: maximum value of the single-factor index. The evaluation results are divided into five grades: $P_{com} \leq 0.7$, safe; $0.7 < P_{com} \leq 1.0$, warning; $1 < P_{com} \leq 2$, light pollution; $2 < P_{com} \leq 3$, moderate pollution; and $P_{com} > 3$, heavy metal pollution.

*2.4. Soil Quality Index Evaluation Method*

2.4.1. Data Sets: Total and Minimum

First, the total data set (TDS) was established. In this study, a total of 18 soil physical, chemical and biological indicators were selected. To eliminate the overlapping information between the primary indicators, a correlation analysis of the evaluation indicators was carried out, and the two indicators with the largest absolute values of the correlation coefficients were selected, leaving the indicators with a weak correlation with other indicators. Second, the minimum data set (MDS) was established. The PCA method was selected for grouping, and the soil evaluation index with a load $\geq 0.5$ in the principal component with a characteristic value $\geq 1$ was selected to be divided into a group. For indicators that may

enter different groups, a group with low correlation was selected [52]. The norm value of the evaluation index is calculated as follows:

$$N_{ik} = \sqrt{\sum_i^k (\mu_{ik}^2 \lambda_k)} \tag{3}$$

where $N_{ik}$ represents the norm value of the first $k$ principal components of the $i$th index whose eigenvalue is greater than 1; $\mu_{ik}$ represents the loading of the $i$th index on the $k$th principal component and $\lambda_k$ is the eigenvalue of the $k$th principal component.

### 2.4.2. Construction of the Soil Quality Scoring Model

After determining the indicators of the TDS and MDS, they were divided into "more is better" and "less is better" according to their role in soil [24]. We used Equation (4) for "more is better" and Equation (5) for "less is better." The data of the selected indicator were mapped to [0.1,1] using Equations (4) and (5).

$$s = 0.1 + \left( \frac{a_i - x_{imin}}{x_{imax} - x_{imin}} \right) * 0.9 \tag{4}$$

$$s = 1.1 - \left[ 0.1 + \left( \frac{a_i - x_{imin}}{x_{imax} - x_{imin}} \right) * 0.9 \right] \tag{5}$$

where $s$ is the normalized score, $a_i$ is the optimal scaling quantification value of the $i^{\text{th}}$ indicator and $x_{imax}$ and $x_{imin}$ are the maximum and minimum values of the $i^{\text{th}}$ indicator, respectively. Combined with the previous research and the actual measurement of indicators, in this study, Formula (5) was used to calculate the score for sand, silt and EC, and Formula (4) was used for the rest of the indicators.

### 2.4.3. Evaluation of Index Weight and SQI Calculation

The SQI is a tool that effectively combines a variety of information to make a multi-objective decision [53]. By calculating the weight and score of each soil quality evaluation index, the index score is integrated into an equation, which is a comprehensive reflection of soil function. The larger its value, the better the soil quality. The calculation formula for the SQI is as follows:

$$\text{SQI} = \sum_{i=1}^n W_i N_i \tag{6}$$

where $W_i$ represents the weight of the $i$th evaluation index, $N_i$ is the membership value of the $i$th evaluation index and $n$ is the number of evaluation indexes. The soil quality is divided into five grades: very high, high, moderate, low and very low (I, II, III, IV and V, respectively) [54].

## 3. Results

### 3.1. Statistical Characteristics Analysis of Soil Properties

The heavy metal contents of the soil are shown in Table 1. The results show that the average content of heavy metals in the soil in the study area followed the order of (mg/kg) Mn (1525.96) > Ni (851.22) > Cr (752.55) > Cu (610.45) > Zn (97.12) > Cd (0.69), all of which exceeded the background value of heavy metals in Xinjiang soil [55], and Ni and Cu exceed by the highest multiples, at 31 times and 21 times, respectively. Taking the national environmental quality standards for soil [56] as the reference criterion, the Ni, Cu, Cr and Cd contents in the study area were found to exceed the standard by 20, 11, 4 and 1 times, respectively. The Zn content in the study area was lower than the national soil environment class II standard. The coefficient of variation, from the largest to smallest, followed the order Ni (55.19%) > Cu (46.17%) > Cd (32.75%) > Cr (26.16%) > Zn (18.05%) > Mn (14.82%). It is worth noting that the coefficient of variation of heavy metals was less

than 100%, or medium variation, indicating that the distribution of heavy metal content in the study area is relatively uniform.

**Table 1.** Statistical characteristics of heavy metal content in soil ($n$ = 30).

| Item | Ni | Cu | Cr | Cd | Zn | Mn |
|---|---|---|---|---|---|---|
| Min (mg/kg) | 337.19 | 289.60 | 431.26 | 0.30 | 70.69 | 1013.83 |
| Max (mg/kg) | 1904.88 | 1195.08 | 1013.83 | 0.99 | 139.46 | 1899.00 |
| Mean (mg/kg) | 851.22 | 610.45 | 752.55 | 0.69 | 97.12 | 1525.96 |
| Standard Deviation | 469.82 | 281.82 | 196.86 | 0.22 | 17.53 | 226.13 |
| Coefficient of Variation (%) | 55.19 | 46.17 | 26.16 | 32.75 | 18.05 | 14.82 |
| BG$_1$ [a] (mg/kg) | 26.60 | 26.70 | 49.3 | 0.12 | 68.80 | 688 |
| BG$_2$ [b] (mg/kg) | 40.00 | 50.00 | 150.00 | 0.30 | 200.00 | - |

a: BG$_1$: background value of heavy metals in Xinjiang soil; b: BG$_2$: national soil environment class II standard.

Table 2 shows the physical, chemical and biological properties and descriptive statistics of the soil in the study area. Among the soil physical evaluation indexes, SWC was low, with an average value of 5.56%. According to the American soil texture classification standard [57], clay, silt and sand in the soil reached values of 6.56%, 79.13% and 14.31%, respectively, and the soil type was silt. In terms of the soil chemical evaluation index, the soil pH value varied from 4.19 to 7.81, with an average value of 6.48, which is weakly acidic. Referring to the nutrient classification standard of the second national soil survey [2], the contents of TP, TK and AK in the soil were found to be at the first level (TP: > 1 g/kg, TK: > 5 g/kg, AK: > 200 mg/kg). The contents of SOM, TN, AN and AP were at the fourth level (SOM: 10–20 g/kg, TN: 0.75–1 g/kg, AN: 60–90 mg/kg, AP: 5–10 mg/kg). In terms of soil biological indicators, the range of URE was 0.06–1.46, with an average of 0.56; the INV range was 0–0.06, and the mean value was 0.02; NPH ranged from 5.30 to 22.22, with an average of 10.66; PPO ranged from 12.64 to 75.64, with an average of 42.76; CAT ranged from 0.01 to 23.41, with an average of 9.49. The analysis of the coefficient of variation of each evaluation index showed that only the coefficient of variation of CAT was greater than 100%, indicating strong variation, and the coefficients of variation of other evaluation indexes were between 13.79% and 96.17%, indicating medium variation.

**Table 2.** Statistical characteristics of soil quality evaluation indexes ($n$ = 30).

| Indicator Type | Indicator | Unit | Min | Max | Mean | SD | CV |
|---|---|---|---|---|---|---|---|
| Soil physical index | SWC | % | 1.00 | 9.00 | 5.56 | 0.03 | 44.98 |
| | Clay | % | 2.00 | 11.00 | 6.56 | 0.02 | 33.45 |
| | Silt | % | 53.00 | 93.00 | 79.13 | 0.11 | 13.79 |
| | Sand | % | 0.01 | 45.00 | 14.31 | 0.12 | 88.11 |
| Soil chemistry indicators | pH | - | 4.19 | 7.81 | 6.48 | 1.20 | 17.77 |
| | EC | mS/cm | 2.50 | 3.37 | 2.83 | 0.29 | 10.25 |
| | SOM | g/kg | 5.71 | 67.97 | 23.11 | 19.33 | 80.77 |
| | TN | g/kg | 0.17 | 3.03 | 0.94 | 0.92 | 94.17 |
| | TP | g/kg | 1.06 | 4.28 | 1.97 | 0.96 | 47.08 |
| | TK | g/kg | 3.85 | 11.94 | 9.17 | 2.63 | 27.72 |
| | AN | mg/kg | 14.21 | 194.34 | 75.52 | 64.52 | 82.43 |
| | AP | mg/kg | 5.08 | 15.38 | 8.24 | 3.01 | 35.32 |
| | AK | mg/kg | 104.60 | 410.10 | 249.63 | 108.95 | 42.56 |
| Soil biometric indicators | URE | mg g$^{-1}$ 24h$^{-1}$ | 0.01 | 1.46 | 0.51 | 0.49 | 94.12 |
| | INV | mg g$^{-1}$ 24h$^{-1}$ | 0.00 | 0.06 | 0.02 | 0.02 | 96.24 |
| | NPH | mg g$^{-1}$ 24h$^{-1}$ | 5.30 | 22.22 | 10.66 | 6.00 | 53.61 |
| | PPO | mg g$^{-1}$ 24h$^{-1}$ | 12.64 | 75.64 | 42.76 | 24.95 | 54.69 |
| | CAT | mg g$^{-1}$ 24h$^{-1}$ | 0.00 | 23.41 | 9.49 | 9.00 | 100.85 |

SWC: soil water content, EC: electrical conductivity, SOM: soil organic matter, TN: total nitrogen, TP: total phosphorus, TK: total potassium, AN: alkali-hydrolyzable nitrogen, AP: available phosphorus, AK: available potassium, URE: urease, INV: invertase, NPH: neutral phosphatase, PPO: polyphenol oxidase, CAT: catalase.

### 3.2. Evaluation of Heavy Metal Pollution in Soil

The single-factor pollution assessment of different heavy metals in soil was carried out according to Formula (1). The evaluation results show (Figure 2) that the single-factor pollution indexes of Ni, Cu, Cr, Cd, Zn and Mn were in the ranges of 12.58–71.72, 10.79–45.22, 8.69–20.91, 2.13–9.22, 1.02–2.03 and 1.77–2.76, respectively. Among them, the average single-factor pollution indexes of heavy metal elements Ni, Cu, Cr and Cd were 32.00, 22.86, 15.26 and 5.71, respectively, all of which are greater than the third level of heavy metal pollution. The mean value of the single-factor pollution index of Mn was 2.22, which is at the moderate pollution level; the average single-factor pollution index value of Zn was 1.41, which is at the level of mild pollution. The Nemerow comprehensive pollution index of heavy metal pollution in the soil was evaluated according to Formula (2). The evaluation results show that the minimum value of the Nemerow comprehensive pollution index of the samples in the study area was 14.26 (>3), indicating that all samples were heavily polluted, and the standard rate of heavy metals was exceeded by 100%. Therefore, necessary ecological restoration measures need to be taken to lessen the heavy metal pollution in the soil around tailings pond.

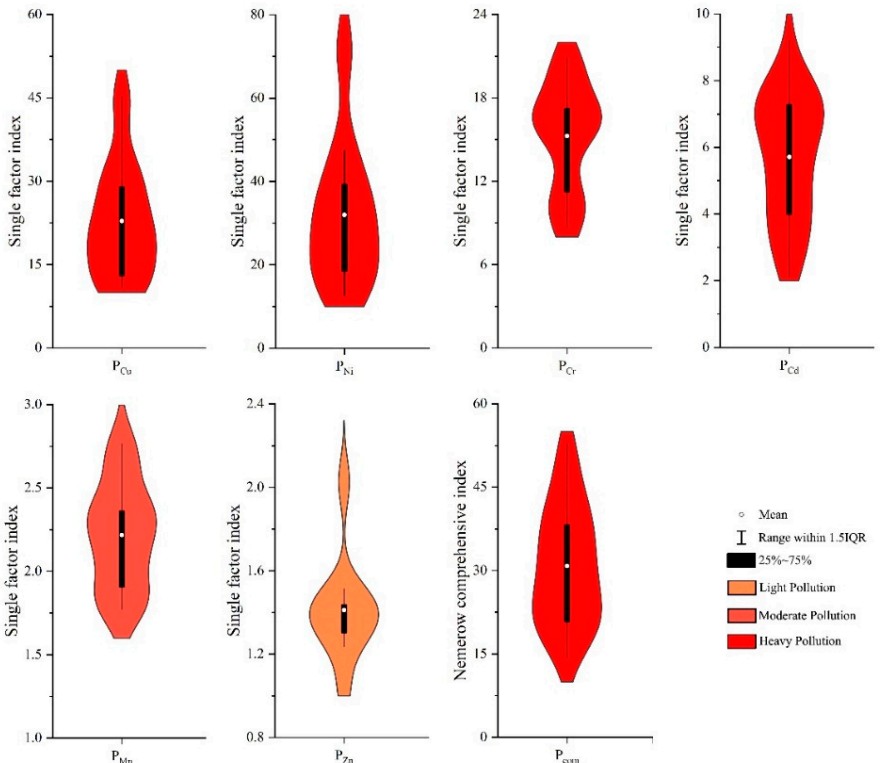

**Figure 2.** Statistical characteristics of soil heavy metal pollution indexes (*n* = 30).

### 3.3. Correlation Analysis of Soil Heavy Metals and Evaluation Indexes

The correlation analysis results of soil heavy metals and evaluation indexes showed that (Figure 3) the correlation between the contents of soil heavy metals in the study area was weak, but the correlation between some heavy metal contents and evaluation indicators was high. Cu had a certain correlation with TK and EC, and the correlation coefficients were −0.89 and 0.69, respectively; Cr was negatively correlated with URE, INV, PPO and silt, and the correlation coefficients were −0.83, −0.78, −0.65 and −0.64, respectively; Mn was positively correlated with PPO, and the correlation coefficient was 0.87; Zn had a high correlation with TP, AP, silt and sand, and the correlation coefficients were 0.88, −0.64, −0.86 and 0.83, respectively. In addition, there was also a certain correlation between soil quality evaluation indicators. SOM was negatively correlated with pH and positively correlated with TN, AN and AP. TN was positively correlated with AN, AP and URE, and

negatively correlated with pH. TP was negatively correlated with SWC, clay and silt, and positively correlated with sand. TK was negatively correlated with EC. AN was positively correlated with INV and negatively correlated with EC. AK was positively correlated with URE, PPO and silt, and negatively correlated with sand. URE was positively correlated with INV and PPO, and negatively correlated with pH. CAT was negatively correlated with EC. SWC had different correlations with different particle sizes, which were positively correlated with clay and silt and negatively correlated with sand. EC was positively correlated with clay. Clay was positively correlated with silt and negatively correlated with sand. Silt was negatively correlated with sand.

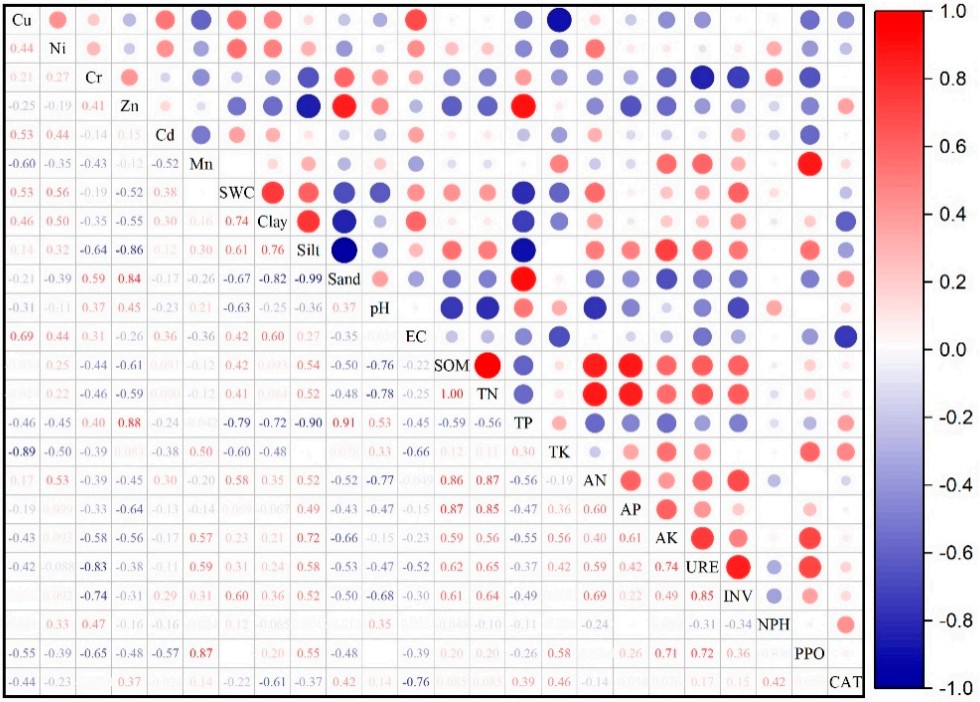

**Figure 3.** Correlation matrix of soil properties (*n* = 30).

*3.4. Soil Quality Evaluation*

3.4.1. Established Minimum Data Set

The principal component load matrix after varimax rotation showed that the five principal components had eigenvalues $\geq$ 1, and the cumulative total interpretation variance was 94.05%, which has a strong interpretation ability (Table 3). In the first principal component (PC1), the soil parameters selected according to the results of load and norm value were SWC, silt, sand, SOM, TN, TP, AN, INV and URE. However, the multivariate correlation between these indicators showed a good correlation (Figure 3), and only the SOM with the highest factor load and the largest norm value was retained in the MDS. Clay is an important soil quality evaluation index, and has a profound impact on the changes in soil quality in desert steppe [58]. In this study, clay had a higher load and higher norm value on PC2, and had a low correlation with URE. Therefore, clay was also added to the MDS. The pH value and PPO were selected in PC3, and the pH value was maintained in the MDS according to norm value results. In PC4, only NPH had a large load, so NPH was selected as input for the MDS. In PC5, AP and CAT had high loads. According to the calculation results of the norm value, only AP was retained to input into the MDS. Therefore, the final soil indicators retained in the MDS were SOM, clay, URE, pH, NPH and AP.

**Table 3.** Principal component analysis results and factor load matrix.

| Indicators | Groups | PC1 | PC2 | PC3 | PC4 | PC5 | Norm |
|---|---|---|---|---|---|---|---|
| SOM | 1 | 0.852 | 0.271 | −0.354 | 0.210 | −0.172 | 2.523 |
| TN | 1 | 0.848 | 0.287 | −0.388 | 0.150 | −0.149 | 2.522 |
| TP | 1 | −0.835 | 0.404 | −0.152 | −0.190 | 0.020 | 2.493 |
| Silt | 1 | 0.837 | −0.227 | 0.436 | 0.014 | −0.041 | 2.475 |
| Sand | 1 | −0.811 | 0.316 | −0.440 | 0.003 | 0.006 | 2.448 |
| AN | 1 | 0.827 | −0.009 | −0.450 | −0.035 | −0.026 | 2.412 |
| SWC | 1 | 0.732 | −0.488 | −0.113 | 0.230 | 0.367 | 2.329 |
| INV | 1 | 0.773 | 0.234 | −0.143 | −0.297 | 0.455 | 2.309 |
| AK | 1 | 0.706 | 0.394 | 0.451 | 0.108 | −0.116 | 2.242 |
| Clay | 2 | 0.498 | −0.685 | 0.363 | −0.091 | 0.169 | 2.045 |
| URE | 2 | 0.749 | 0.506 | 0.132 | −0.307 | 0.231 | 2.380 |
| EC | 2 | 0.059 | −0.921 | 0.072 | 0.057 | −0.238 | 1.878 |
| TK | 2 | −0.005 | 0.847 | 0.409 | 0.003 | −0.239 | 1.829 |
| pH | 3 | −0.723 | 0.079 | 0.589 | 0.149 | −0.062 | 2.226 |
| PPO | 3 | 0.417 | 0.460 | 0.659 | −0.191 | 0.058 | 1.810 |
| NPH | 4 | −0.128 | −0.048 | 0.234 | 0.922 | 0.224 | 1.273 |
| AP | 5 | 0.666 | 0.327 | −0.142 | 0.286 | −0.579 | 2.118 |
| CAT | 5 | −0.177 | 0.699 | −0.124 | 0.412 | 0.500 | 1.678 |
| Eigenvalue | | 7.817 | 4.024 | 2.316 | 1.523 | 1.248 | |
| Initial eigenvalue | | 43.43% | 22.36% | 12.87% | 8.46% | 6.93% | |
| Cumulative contribution | | 43.43% | 65.79% | 78.66% | 87.12% | 94.05% | |

### 3.4.2. Applicability Verification of the Soil Quality Evaluation Method Based on the Minimum Data Set

Through the statistical analysis, the indicator data set can be simplified, but it will lead to a decrease in evaluation accuracy. Therefore, it is necessary to verify the applicability of the MDS of evaluation indicators for a specific area or a specific soil. The common factor variance of each index in the total data set (TDS) and the MDS was obtained through PCA, and then the weight of each index was obtained (Table 4). The indicators were standardized using Formulas (4) and (5), and then substituted in Formula (6) to calculate the SQI of different data sets. Among them, the TDS SQI (SQI-TDS) was between 0.16 and 0.70, with an average value of 0.42; the MDS SQI (SQI-MDS) was between 0.17 and 0.61, with an average of 0.39. The results of soil quality classification using the TDS and MDS methods (Table 5) show that the samples with an SQI at grades III, IV and V calculated based on the TDS method accounted for 3.33%, 13.33% and 83.33% of the total samples, respectively, whereas the samples with an SQI at grades IV and V calculated based on the MDS method accounted for 3.33% and 96.67% of the total samples, respectively. From the classification results, the classification results of the SQI-TDS and SQI-MDS are basically the same, indicating that the SQI calculated based on the MDS has a certain degree of reliability. In addition, the SQI-TDS was positively correlated with the SQI-MDS (Figure 4), and the linear fitting equation was $y = 0.79x + 0.06$ ($R^2 = 0.76$). Therefore, the MDS used in this study can better reflect the soil quality information represented by TDS.

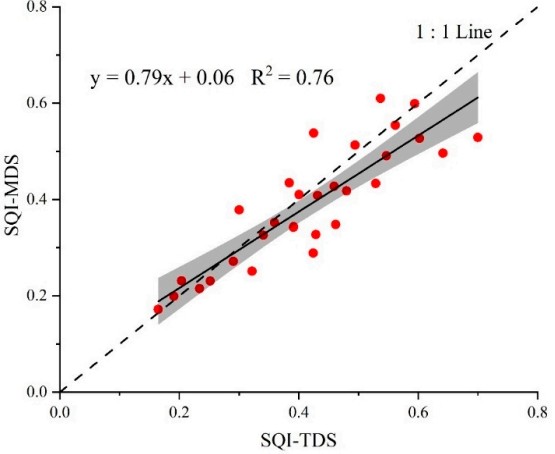

**Figure 4.** Correlation between soil quality index of the minimum data set and the full data set ($n = 30$).

**Table 4.** Commonality and weight of the TDS and MDS.

| Indicators | Groups | TDS | | MDS | |
|---|---|---|---|---|---|
| | | Communality | Weight | Communality | Weight |
| SOM | 1 | 0.998 | 0.059 | 0.983 | 0.194 |
| TN | 1 | 0.997 | 0.059 | | |
| TP | 1 | 0.920 | 0.054 | | |
| Silt | 1 | 0.944 | 0.056 | | |
| Sand | 1 | 0.951 | 0.056 | | |
| AN | 1 | 0.888 | 0.052 | | |
| SWC | 1 | 0.974 | 0.058 | | |
| INV | 1 | 0.968 | 0.057 | | |
| AK | 1 | 0.882 | 0.052 | | |
| Clay | 2 | 0.886 | 0.052 | 0.972 | 0.192 |
| URE | 2 | 0.982 | 0.058 | 0.612 | 0.121 |
| EC | 2 | 0.918 | 0.054 | | |
| TK | 2 | 0.942 | 0.056 | | |
| pH | 3 | 0.902 | 0.053 | 0.728 | 0.144 |
| PPO | 3 | 0.860 | 0.051 | | |
| NPH | 4 | 0.974 | 0.058 | 0.928 | 0.183 |
| AP | 5 | 0.988 | 0.058 | 0.850 | 0.168 |
| CAT | 5 | 0.954 | 0.056 | | |

**Table 5.** Classification and proportion of soil quality grades using different analysis methods.

| SQI | Rate | Soil Quality Grades | | | | |
|---|---|---|---|---|---|---|
| | | Very High | High | Moderate | Low | Very Low |
| SQI-TDS | % | >0.85 | 0.75–0.85 | 0.65–0.75 | 0.55–0.65 | <0.55 |
| | | 0 | 0 | 3.33 | 13.33 | 83.34 |
| SQI-MDS | % | >0.87 | 0.78–0.87 | 0.69–0.78 | 0.60–0.69 | <0.60 |
| | | 0 | 0 | 0 | 3.33 | 96.67 |

## 4. Discussion

### 4.1. Analysis of Heavy Metal Pollution Sources in Soil

The source analysis of soil heavy metals is the basis of soil heavy metal pollution prevention and control. Clarifying the source of soil heavy metal pollutants is of great significance to carrying out targeted heavy metal pollution control. In this study, the positive matrix factorization (PMF) model was used to analyze the sources of heavy metal pollution in the soil of the study area. The results show (Figure 5) that Factor 1 contributed 63.0% and 71.8% of the heavy metals Cu and Ni, respectively. Combined with the single-factor pollution evaluation index, the pollution degree of heavy metals Cu and Ni was the highest, at 22.86 and 32.00, respectively. There was a significant amount of Cu and Ni residue in copper-nickel mine tailings. Under the action of AMD, Cu and Ni continue to diffuse into the soil, resulting in serious Cu and Ni pollution in the study area. Therefore, Factor 1 may come from industrial production. The contribution of Factor 2 to Mn was 54.5%. As we all know, Mn is naturally abundant in soil, and its content in surface soil is not significantly altered by human activities [59]. Relevant studies also show that the content of Mn in soil usually depends on the soil's parent material [60–62]. In this study, the minimum coefficient of variation of Mn was 14.82%, which is close to the weak variation value (10%), indicating that it is evenly distributed in space and is less affected by human factors. Therefore, Factor 2 may come from natural sources. The contribution of Factor 3 to Cr, Zn and Cd was 55.6%, 44.3% and 41.6%, respectively. Diana et al. [63] showed that Cr, Cd and Zn mainly come from a mix of industrial production and transportation sources. Fuyun County has a developed mining industry and is an important producer of metal and nonmetallic minerals in the country. Ore mining, smelting and coal combustion

are significant contributors of Cr, Cd and Zn in soil [64,65]. At the same time, the study area is located near national and provincial roads, and the pollution caused by vehicle driving also plays an important role in the accumulation of Cr, Cd and Zn in soil [66,67]. Therefore, Factor 3 may come from the load pollution source of industrial production and traffic pollution.

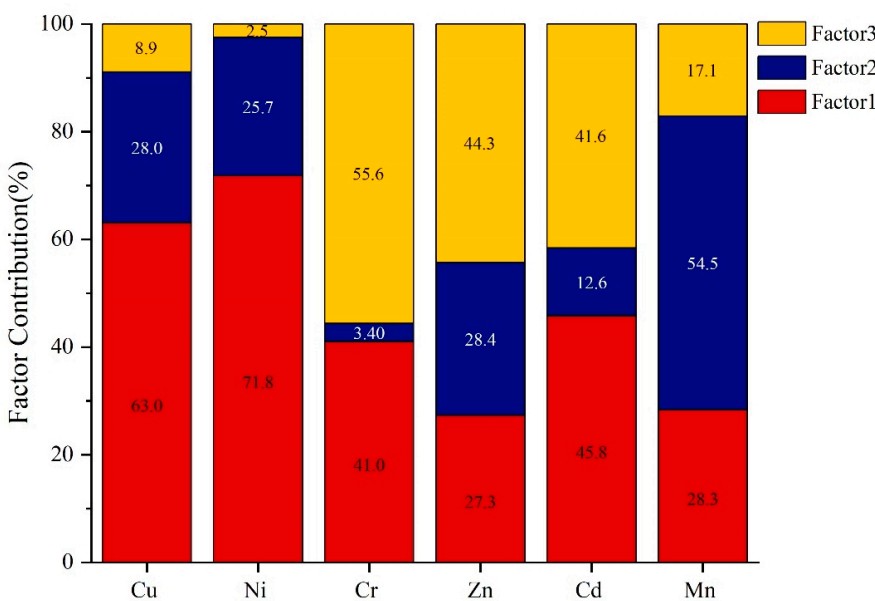

**Figure 5.** Analytical contribution of heavy metal PMF source.

### 4.2. Establishing the Minimum Data Set for Soil Quality Evaluation

In general, additional indicators can improve the comprehensive analysis of soil quality and improve the accuracy of evaluation; however, because of the large number of indicators, complex experimental analysis is time-consuming and laborious [54]. Therefore, it is necessary to use statistical analysis for specific soils or regions to delete indicators and build the MDS for soil quality evaluation [68,69]. In this study, 18 soil parameters were selected as soil quality evaluation indicators, and the sample data were screened using PCA, correlation analysis and norm values. The final MDS contained SOM, clay, AP, pH, URE and NPH, and the index screening rate was 66.67%. By considering the physical, chemical and biological characteristics of soil, the evaluation index system was effectively simplified, and the impact of overlapping information between evaluation indexes on the evaluation results was minimized. SOM directly affects the physical, chemical and biological characteristics of soil [70], and can play a key role in the preservation and release of nutrients. Therefore, SOM is the most important indicator for soil quality evaluation and the most common indicator in the MDS [71–73] In our study, SOM had the highest load value in PC1 and the highest norm value (Table 4), indicating that SOM has a great impact on the soil quality of the study area, so it was added to the MDS. Soil texture (clay, silt and sand) is crucial to determining soil quality because it provides an isolated microhabitat for microorganisms, thereby increasing the diversity and richness of microorganisms [74]. Among them, clay particles are significantly related to soil quality and have a great impact on soil nutrient cycling [75]. Jin et al. [19] showed that clay is often selected for the MDS in the process of evaluating the soil quality of sloping farmland, and its frequency of use reached 79%. AP is one of the most restrictive factors for plant growth and plays a crucial role in the net carbon absorption of ecosystems [31]. The content of soil AP in the study area was low, at the fourth level, indicating that AP imposes great restrictions on the soil quality of the study area. Soil pH directly reflects the occurrence of soil chemical reactions and affects the availability of nutrients required for plant growth [76]. Affected by AMD, the soil in the study area was acidic, with a low pH of 4.19. However, desert plants in

Xinjiang are adapted to an alkaline environment. Studies have shown that desert plant diversity is positively correlated with pH [77,78], so low pH will not be conducive to plant growth in the study area, and will further restrict the improvement in soil quality. Based on the establishment of the MDS, and by collecting a large number of research results for soil quality evaluation, the use frequencies of different indicators in the MDS were compared (Figure 6). The use frequencies of SOM, pH and AP were higher, namely 55.81%, 33.72% and 33.72%, respectively; the frequencies of clay, TN, TK, TP, AN and AK were 19.77%, 24.42%, 16.28%, 17.44%, 16.28% and 22.09%, respectively; SWC, silt and sand were used less frequently, at 4.65%, 4.65% and 8.14%, respectively. The production rate of soil enzymes is affected by environmental effects and ecological interactions. Enzyme activity is sensitive to small changes in soil and is a sensitive indicator of soil ecological stress [79]. Therefore, soil enzyme activity was included in the soil quality evaluation index system for assessing heavy metal pollution stress. In this study, URE, INV, NPH, PPO and CAT were used as soil quality evaluation indicators. Among them, URE and NPH had large load values on PC2 and PC4 (Table 4), and they were finally included in the MDS. Wang et al. [80] also used urease and phosphatase in the MDS in the process of evaluating the soil quality of iron tailings wasteland. Tian et al. [52] selected urease and catalase for inclusion in the MDS in the process of evaluating the soil quality of desert steppe. In addition, the correlation analysis results show that there was a positive correlation between the MDS and TDS ($R^2$ = 0.76), so the selection of MDS can well reflect the soil quality in the study area.

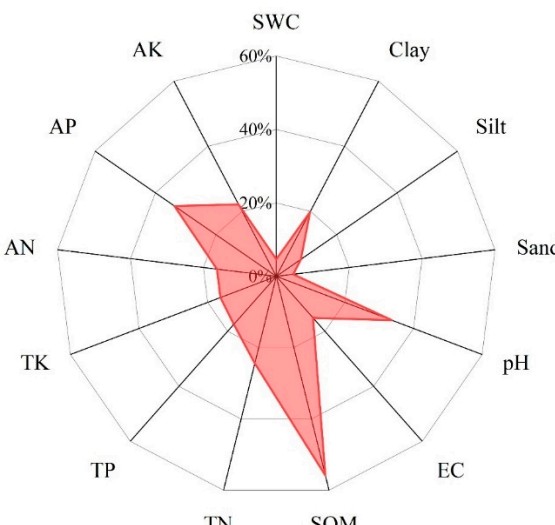

**Figure 6.** Usage frequency of different indicators in the MDS.

### 4.3. Impact of Heavy Metal Pollution on Soil Quality

Heavy metal pollution has existed in soil for a long time, seriously affecting soil quality [11]. An increase in the content of heavy metals in soil will reduce the content of soil nutrients [10], affect the availability of nutrients in soil solution and soil enzyme activity and eventually lead to a sharp decline in vegetation and land productivity [81,82]. The contents of SOM, TN, AN and AP in the study area were at the fourth level, and the soil quality was poor. The correlation analysis results show that the heavy metals Ni and Cu in the soil were negatively correlated with SOM, TN, AN, AP, AK, etc. Therefore, the accumulation of heavy metals in the soil has an adverse impact on the content of soil nutrients. Among many evaluation indicators, soil enzyme activity is the most sensitive indicator of heavy metals, and also an important indicator of soil quality and health, because it is directly related to soil carbon, nitrogen and phosphorus cycles [21,83,84]. Heavy metal ions in soil may compound with substrates, combine with protein-active groups of enzymes or react with enzyme-substrate complexes, thereby affecting enzyme activity [83]. However, the response of enzyme activity to heavy metal ions is relatively complex. In the same

heavy-metal-polluted environment, different heavy metals have different effects on enzyme activity, and the same heavy metal has different effects on different enzymes [83,85,86]. The results of this study show that Cr was negatively correlated with URE, INV and PPO, whereas Mn was positively correlated with PPO. Yang et al. [87] also show that Cr is negatively correlated with a variety of enzymes, including URE, whereas PPO is positively correlated with Cd, Zn, Cu and other heavy metals. In addition, the analysis results for soil quality and heavy metal pollution degree show that (Figure 7) there was a negative correlation between soil quality and heavy metal pollution degree. The regression equation between the SQI-MDS and $P_{com}$ was $y = -0.014x + 0.88$ ($R^2 = 0.75$), and the regression equation between the SQITDS and $P_{com}$ was $y = -0.012x + 0.81$ ($R^2 = 0.64$). Therefore, the heavy metal soil pollution caused by the tailings pond will lead to a decrease in soil quality, and the SQI-MDS can effectively reflect the degree of soil heavy metal pollution.

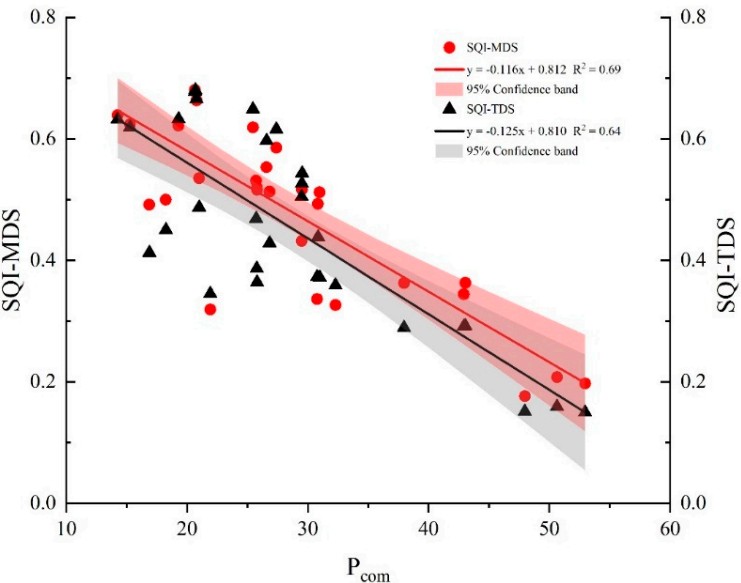

**Figure 7.** Correlation between SQI and $P_{com}$ ($n = 30$).

## 5. Conclusions

Tailings accumulation not only causes serious soil heavy metal pollution but also has a negative impact on soil quality. The results of the Nemerow comprehensive pollution assessment show that the heavy metal pollution in the soil in the study area was at a high level of heavy metal pollution. The single-factor pollution evaluation results show that Cu, Ni, Cr and Cd are heavily polluting; Mn is moderately polluting; Zn is slightly polluting. The results of the PMF model show that Cu and Ni come from industrial sources; Cr, Cd and Zn come from a mix of industrial production and transportation sources; Mn comes from natural sources. Through PCA of 18 soil characteristic indexes, combined with correlation analysis and norm values, six soil indexes were finally selected as the MDS for soil quality evaluation, namely, clay, pH, SOM, AP, URE and NPH. The fitting results of the SQI-TDS and SQI-MDS show that there was a positive correlation between the SQI-TDS and SQI-MDS ($R^2 = 0.76$), the average values of the two evaluation results were 0.42 and 0.39, respectively, and the soil quality conditions were very poor. The fitting results of the SQI and $P_{com}$ calculated with different data sets show that the SQI and $P_{com}$ were negatively correlated, and the SQI calculated based on the MDS had a higher correlation with the Nemerow index ($R^2 = 0.69$), indicating that the SQI calculated based on the MDS can better reflect the degree of heavy metal pollution in soil. In general, our results show that the leakage of AMD from tailings accumulation has a negative impact on the soil quality of desert steppe. The MDS constructed in this study can reflect the soil quality and heavy metal pollution at the same time.

**Author Contributions:** J.S.: Conceptualization, Methodology, Software, Investigation, Formal Analysis, Writing—Original Draft, W.Q.: Methodology, Sample analysis, Z.Z.: Investigation, Conceptualization, Writing—Review and Editing, Z.J.: Funding Acquisition, Conceptualization, Methodology, Writing—Review and Editing; X.X.: Funding Acquisition, Conceptualization, Methodology, Writing—Review and Editing. All authors have read and agreed to the published version of the manuscript.

**Funding:** The National Key Research and Development Program of China (2018YFC1802903).

**Institutional Review Board Statement:** Not applicable.

**Informed Consent Statement:** Not applicable.

**Data Availability Statement:** The data sets generated and/or analyzed during the current study are available from the corresponding author on reasonable request.

**Conflicts of Interest:** The authors declare no conflict of interest.

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
