# Peer review of "Influence of Acid Mine Drainage Leakage from Tailings Ponds on the Soil Quality of Desert Steppe in the Northwest Arid Region of China"

_land, doi:10.3390/land12020467_

Round 1
Reviewer 1 Report
The article makes a good impression.
The data obtained is well analyzed in depth.
However, there are small questions that are recommended to be answered.
2.2. Analysis of soil samples
What method was used to extract heavy metals from the soil?
The recovery extent of each element was between 92% and 104%.
How did you get 104%?
2.3.1. Single-factor pollution index method
Why is Xinjiang district chosen as the standart territory?
There are 2 types of soils on your research area. "The soil types around the study area are mainly chestnut soil and brown calcium soil."
You have chosen soil in Xinjiang as a reference for pollution assessment. What are these soils called by classification? Are these soils similar in their properties to the soils of the research area? If not, then they cannot be considered a standart.
It would be interesting to measure the filtration rate, since soils of light granulomeric composition predominate in the research area, which means that contamination of this territory is really very dangerous.
Is there any data on the concentration of heavy metals in plants on your territory? If so, it could be added.
Author Response
Thank you for your valuable comments. In accordance with your comments and suggestions, this paper is revised as follows.
Point 1: What method was used to extract heavy metals from the soil?
Response 1: Thanks for your comment. We use the microwave digestion method to extract heavy metals from soil, which is described in detail in the new version.
Point 2: The recovery extent of each element was between 92% and 104%. How did you get 104%?
Response 2: Thanks for your comment. Standard recovery is an important method for laboratory quality control. We added quantitative standards to the blank sample matrix and followed the sample processing steps to obtain the recoveries. The calculation formula is as follows:
Usually, a recovery rate between 80% and 120% indicates that the experiment is satisfactory. Therefore, 104% is the ratio of the measured value to the theoretical value.
Point 3: Single-factor pollution index method. Why is Xinjiang district chosen as the standart territory?
Response 3: Thanks for your comment. The content of heavy metals in soils developed by different parent material types varies greatly, resulting in a strong heterogeneity in spatial distribution of heavy metals, based on which we chose the soil background values of the administrative region where the study area is located as the evaluation criteria.
Point 4: There are 2 types of soils on your research area. "The soil types around the study area are mainly chestnut soil and brown calcium soil." You have chosen soil in Xinjiang as a reference for pollution assessment. What are these soils called by classification? Are these soils similar in their properties to the soils of the research area? If not, then they cannot be considered a standart.
Response 4: Thanks for your comment. There are two main evaluation standards chosen for the evaluation of soil heavy metal pollution, one is the national soil environmental quality standard, and the other is the regional soil heavy metal background value. Due to the vast size of China and the obvious differences in heavy metal content in different regions, the background value of soil in Xinjiang was chosen as the evaluation standard in order to make the evaluation results closer to the actual local situation. It is worth mentioning that there are many types of soils in Xinjiang. According to the results of the second soil census in Xinjiang, the soils in Xinjiang can be divided into 7 soil classes and 32 soil types. Due to the wide variety of soil types, following the background values of different soil types as evaluation criteria may lead to confusion in the evaluation results of regional soil heavy metal pollution and hinder the development of regional soil pollution treatment. However, if a systematic set of heavy metal pollution evaluation criteria is established according to soil types in the future, this may further improve the accuracy and science of soil environmental quality evaluation, which may require more work by scholars.
Point 5: It would be interesting to measure the filtration rate, since soils of light granulomeric composition predominate in the research area, which means that contamination of this territory is really very dangerous. Is there any data on the concentration of heavy metals in plants on your territory? If so, it could be added.
Response 5: Thank you for your comments. It is very unfortunate that we do not have obtained data related to the heavy metal content of plants.

Reviewer 2 Report
land-2154288-review
Influence of acid mine drainage leakage from tailings ponds on the soil quality of desert steppe in the Northwest Arid Region
In order to understand the impact of tailings stockpiles on the soil quality of desert steppe, this study analyzed 18 indicators in the sample and analyzed the soil-quality status of desert steppe based on the soil-quality index (SQI) and Nemerow pollution index (Pcom). This is a very interesting and meaningful research topic. Some suggestions and comments are as follows:
1. In the introduction, the logic and structure are relatively clear. It is suggested to add several case data to illustrate the urgency and particularity of soil quality degradation caused by acid mine drainage leakage from tailings ponds. The leakage of tailings pond is generally treated by curtain grouting. Several references may enlighten you, such as, https://doi.org/10.3390/w14244093, https://doi.org/10.1134/S1064229322010100.
2. In the 2.1. Site description and soil sampling, I think the author should briefly explain the leakage mechanism and path of the tailings pond studied, as well as the flow direction of groundwater. This helps the reader to judge the rationality and validity of the sampling location.
3. The geological and hydrogeological conditions around the studied tailings pond are not clearly described. The process parameters and more details of tests should also be further supplemented.
4. Lines 202-206, 2.5. Data processing and statistical analysis, the description in this part is redundant and has no profound scientific significance. It is recommended to delete.
5. Lines 212, background value of heavy metals in Xinjiang soil, this is an important reference, I think some control soil samples should be tested and comparatively analyzed around the studied area, not Xinjiang soil.
6. In the conclusion, Lines 463-465, In general, our results show that the leakage of AMD from tailings accumulation has a negative impact on the soil quality of desert steppe. The MDS constructed in this study can reflect the soil quality and heavy-metal pollution at the same time. This seems to be a common sense conclusion. I think the author's analysis should be more profound to reveal how heavy metals affect soil quality from the perspective of internal mechanism.
7. The paper title, Influence of acid mine drainage leakage from tailings ponds on the soil quality of desert steppe in the Northwest Arid Region. The author has done some meaningful work, but the study area is desert steppe in the Northwest Arid Region. What is the difference between the results obtained in plain and mountain areas.
8. The language needs to be carefully checked and polished. There are some grammatical errors.
Author Response
Thank you for your valuable comments. In accordance with your comments and suggestions, this paper is revised as follows.
Point 1: In the introduction, the logic and structure are relatively clear. It is suggested to add several case data to illustrate the urgency and particularity of soil quality degradation caused by acid mine drainage leakage from tailings ponds. The leakage of tailings pond is generally treated by curtain grouting.
Response 1: Thank you for your comments and for the recommended articles. We have added to the manuscript about the treatment of tailings pond leaks and have cited the relevant references.
Point 2: In the 2.1. Site description and soil sampling, I think the author should briefly explain the leakage mechanism and path of the tailings pond studied, as well as the flow direction of groundwater. This helps the reader to judge the rationality and validity of the sampling location.
Response 2: Thank you for your comments. We have given a brief description of the formation of acid mine drainage in the new version. In addition, we regret that we have not monitored the groundwater resources around the tailing ponds to provide information on groundwater flow direction. However, in arid areas where underground water level is deep, the surface soil contamination is mainly surface runoff formed by AMD.
Point 3: The geological and hydrogeological conditions around the studied tailings pond are not clearly described. The process parameters and more details of tests should also be further supplemented.
Response 3: Thank you for your comments. In the new version we add a description of the geological and hydrological conditions of the study area.
Point 4: Lines 202-206, 2.5. Data processing and statistical analysis, the description in this part is redundant and has no profound scientific significance. It is recommended to delete.
Response 4: Thank you for your comments. We have deleted the data processing and statistical analysis in the new version
Point 5: Lines 212, background value of heavy metals in Xinjiang soil, this is an important reference, I think some control soil samples should be tested and comparatively analyzed around the studied area, not Xinjiang soil.
Response 5: Thank you for your comments. Choosing reasonable heavy metal pollution evaluation standards is an important prerequisite for improving the scientific and accurate evaluation of regional soil heavy metal pollution risks. Usually, administrative district soil background values or national soil environmental quality standards are chosen in the process of heavy metal pollution evaluation. Due to the vast area of China and the obvious differences in heavy metal contents in different regions, the background values of soil heavy metal contents in Xinjiang were chosen as the evaluation criteria in order to make the evaluation results closer to the actual situation in Xinjiang. It is important to select the heavy metal content of surrounding soil samples as ecological risk evaluation criteria, but this may require extensive soil sample collection to determine regional soil heavy metal geochemical baseline values and thus accurately evaluate the regional heavy metal pollution status. In the future we will focus our research efforts on this aspect.
Point 6: In the conclusion, Lines 463-465, In general, our results show that the leakage of AMD from tailings accumulation has a negative impact on the soil quality of desert steppe. The MDS constructed in this study can reflect the soil quality and heavy-metal pollution at the same time. This seems to be a common sense conclusion. I think the author's analysis should be more profound to reveal how heavy metals affect soil quality from the perspective of internal mechanism.
Response 6: Thank you for your comments. In the discussion section we briefly analyzed the effects of heavy metals on soil nutrients and enzymatic activities in the context of previous research results. However, we may need to conduct an in-depth indoor experimental analysis on the mechanism of heavy metal pollution on soil quality indicators. This may be the next step of our research.
Point 7: The paper title, Influence of acid mine drainage leakage from tailings ponds on the soil quality of desert steppe in the Northwest Arid Region. The author has done some meaningful work, but the study area is desert steppe in the Northwest Arid Region. What is the difference between the results obtained in plain and mountain areas.
Response 7: Thank you for your comments. From the research results of existing articles, no similar studies for plains or mountainous areas have been found. Since this study area is located in the desert grassland of the northwest arid zone, which is more restricted by moisture conditions, soil heavy gold contamination may lead to further deterioration of soil quality. If the study area is located in the southern plains or mountains, the higher precipitation may help the recovery of regional vegetation as well as the leaching of heavy metals, which can help a lot in maintaining or restoring soil quality. Therefore, we speculate that the conclusion of this study may differ somewhat from the southern plains or mountains. If there is an opportunity in the future we would like to conduct a comparative study, which may provide some useful information for regional soil science and environmental science.
Point 8: The language needs to be carefully checked and polished. There are some grammatical errors.
Response 8: Thank you for your comments. We have polished the new manuscript for the second time.

Reviewer 3 Report
The paper is prepared according template and well organized. The Introduction should be completed with several references that were published in the period 2021-2023.
Author Response
Thank you for your valuable comments. In accordance with your comments and suggestions, this paper is revised as follows.
Point 1: The paper is prepared according template and well organized. The Introduction should be completed with several references that were published in the period 2021-2023.
Response 1: Thanks to your comments, we have adjusted some of the references and selected references that are up to date.

Round 2
Reviewer 2 Report
Can be accepted in present form
Author Response
Please find attached the corrected version of our manuscript “Influence of acid mine drainage leakage from tailings ponds on the soil quality of desert steppe in the Northwest Arid Region”. We appreciate the constructive comments and suggestions from reviewers. These opinions help to improve academic rigor if our article. Based on their suggestion and request, we have made corrected modifications on the revised manuscript. We hope that our work can be improved again.
